# Structural Insights into the Distortion of the Ribosomal Small Subunit at Different Magnesium Concentrations

**DOI:** 10.3390/biom13030566

**Published:** 2023-03-20

**Authors:** Ting Yu, Junyi Jiang, Qianxi Yu, Xin Li, Fuxing Zeng

**Affiliations:** Department of Systems Biology, School of Life Sciences, Southern University of Science and Technology, No. 1088 Xueyuan Avenue, Shenzhen 518055, China

**Keywords:** structural distortion, magnesium concentration, ribosome, CryoEM

## Abstract

Magnesium ions are abundant and play indispensable functions in the ribosome. A decrease in Mg^2+^ concentration causes 70S ribosome dissociation and subsequent unfolding. Structural distortion at low Mg^2+^ concentrations has been observed in an immature pre50S, while the structural changes in mature subunits have not yet been studied. Here, we purified the 30S subunits of *E. coli* cells under various Mg^2+^ concentrations and analyzed their structural distortion by cryo-electron microscopy. Upon systematically interrogating the structural heterogeneity within the 1 mM Mg^2+^ dataset, we observed 30S particles with different levels of structural distortion in the decoding center, h17, and the 30S head. Our model showed that, when the Mg^2+^ concentration decreases, the decoding center distorts, starting from h44 and followed by the shifting of h18 and h27, as well as the dissociation of ribosomal protein S12. Mg^2+^ deficiency also eliminates the interactions between h17, h10, h15, and S16, resulting in the movement of h17 towards the tip of h6. More flexible structures were observed in the 30S head and platform, showing high variability in these regions. In summary, the structures resolved here showed several prominent distortion events in the decoding center and h17. The requirement for Mg^2+^ in ribosomes suggests that the conformational changes reported here are likely shared due to a lack of cellular Mg^2+^ in all domains of life.

## 1. Introduction

Metal ions are the second most abundant component after water molecules in living cells and are involved in all fundamental biological processes, including protein synthesis, enzymatic reactions, and others [1]. Protein synthesis is mediated by ribosomes, in which the information carried by mRNA is translated into amino acid sequences. A ribosome requires metal ions, including Mg^2+^, Zn^2+^, and K^+^, to maintain its structure and activity. Mg^2+^ is the most abundant multivalent cation in cells and plays an essential role in the assembly of ribosomes by neutralizing negative charges from phosphates present in the rRNA backbone and enabling the correct folding and compaction of rRNA [2,3]. 

The bacterial 70S ribosome is a complex macromolecule composed of small (30S) and large (50S) subunits. Recently, a high-resolution ribosome structure was determined, showing that a single ribosome in *Escherichia coli* (*E. coli*) contains at least 309 Mg^2+^ ions [4]. It has been known for decades that the structure and function of ribosomes are strongly influenced by the presence of Mg^2+^ [5]. For example, the in vitro association between small and large ribosomal subunits required to form intact ribosomes depends strongly on the Mg^2+^ concentration [6], and decreasing the Mg^2+^ concentration below 1 mM causes the dissociation and subsequent unfolding of 70S ribosomes [2,7,8]. Meanwhile, extremely low Mg^2+^ causes irreversible structural distortions and even disassembly into individual ribosomal constituents [3]. Research on ribosome unfolding showed that EDTA-dialysis could be used to progressively remove the Mg^2+^ from ribosomes and resulted in the conversion of 50S subunits into 21S particles via a 36S intermediate and the conversion of the 30S subunit into 16S particles via a 26S intermediate, in which the 36S and 26S particles were reversible through the readdition of Mg^2+^, whereas the 21S and 16S particles containing only 23S and 16S rRNA were irreversible [2]. The growth of *E. coli* cells under conditions of Mg^2+^ starvation results in ribosome degradation [9]. Furthermore, Mg^2+^ stabilizes the codon–anticodon interaction at the A site and influences the binding of RRF to the ribosome [10,11]. In addition, Mg^2+^ can partly complement the functions of several ribosomal proteins, such as L1, L23, and L34 [6,12]. For instance, an increased Mg^2+^ concentration suppresses the defects in 70S ribosome formation caused by a lack of ribosomal protein L34. Previous studies have also suggested that lower Mg^2+^ concentrations greatly increase the susceptibility of ribosomes to attack by ribonuclease [13,14]. In *E. coli* cells, the concentration of intracellular free Mg^2+^ ranges from 1 mM to 5 mM [15,16]. Although Mg^2+^ is essential for ribosomes, excess Mg^2+^ reduces their translation activity and accuracy [17,18]. For instance, the error frequency measured in vitro at 10 mM Mg^2+^ is 10 times higher than that at 5 mM Mg^2+^ [19]. 

In order to examine the impact of low Mg^2+^ exposure on the 30S structure, we used cryo-electron microscopy (cryo-EM) to explore the structures of 30S particles and characterize a series of unnatural 30S structures at 1 mM Mg^2+^. The 30S ribosomal subunit is composed of one rRNA molecule (16S rRNA) and approximately 21 r-proteins, which are organized into four distinct structural domains: the body (5′ domain), the platform (central domain), the head (3′ major domain), and helix 44 with h45 (3′ minor domain) [20]. The assembly of the 30S subunit is a robust process proceeding via multiple redundant parallel pathways, where the 5′ body domain forms first, followed by the central platform, head domains, and lastly, the 3′ minor domain with the functionally important decoding center [21,22,23,24]. Although ribosome unfolding has been studied with many different techniques, such as the measure of the sedimentation coefficient, viscosity, and diffusion constant, as well as ultraviolet absorption and laser Raman spectroscopy, and despite the fact that evidence has been presented to show that discrete intermediates exist in the unfolding reaction, little information about specific structural changes during unfolding has been provided [3,5]. A recent study of rRNA self-folding showed that in the absence of Mg^2+^ or with Mg^2+^ of up to ~1 mM, the tertiary interactions of the 16S central domain are disrupted, resulting in expanded conformations containing only secondary structures [8]. How does Mg^2+^ facilitate the folding of rRNA fragments, especially the formation of tertiary contacts? In this study, our cryoEM structures showed that the 30S particles collected at 1 mM showed several missing structural features and conformation changes, including missing h44 and S12 and the movement of h17 and h27. These structural distortions in unfolding intermediates provide insight into ribosome biogenesis and should be taken into account in vitro assembling studies. 

## 2. Materials and Methods

### 2.1. E. coli Strains and Cell Culture

In this study, 30S particles were purified from an *E. coli* strain with the mutant DbpA protein overexpressed, obtained from another project conducted by our research group. Since the structural distortion observed in these 30S_DbpA_ at 1 mM Mg^2+^ was shown to be exactly the same as that of 30S purified from the wild-type BL21 strain (data not shown), the cryoEM data of 30S_DbpA_ were used for reconstruction and designated as 30S_1mM_ in this study. Wild-type *E. coli* BL21 and MRE600 were used for the sucrose gradients and purification of 30S at 2.5 and 10 mM Mg^2+^. 

For the sucrose gradient and 30S purification, *E. coli* cells of the strains BL21 and MRE600 were grown to an OD_600_ between 0.6 and 0.8 at 37 °C and 220 rpm in LB medium before harvesting. Yeast cells of the strain BY4742 were grown to an OD_600_ of 0.65 at 30 °C and 220 rpm in YPD medium. HEK293F cells were grown in suspension culture to approximately 2 × 10^6^ cells per milliliter before harvesting. 

### 2.2. Sucrose Gradient Centrifugation Analysis

For the sucrose gradient analysis, 50 mL of cells were collected by centrifugation at 3500 rpm at 4 °C for 10 min and were resuspended in 200 μL of lysis buffer (20 mM HEPES-KOH pH 7.5, 150 mM NH_4_Cl, 4 mM β-mercaptoethanol) with different concentrations of Mg^2+^ and EDTA, according to the experimental requirements, and DNase I (RNase-free) was added at a final concentration of 20 U/mL. The cells were broken with a grinder, followed by centrifugation at 14,000 rpm at 4 °C for 20 min. Fifteen units of A_260_ of clarified lysates were loaded onto a twelve milliliters of sucrose gradient.

For the EDTA-treated assay, the clarified lysates were loaded onto a 10–50% sucrose gradient in lysis buffer with 10 mM EDTA and then centrifuged at 4 °C with an SW41 rotor for 4 h at 35,000 rpm. For the assay of different concentrations of Mg^2+^, the clarified lysates were loaded onto a 10–50% sucrose gradient in lysis buffer with 0.5 mM, 1 mM, 2.5 mM, 5 mM, 10 mM, and 20 mM Mg^2+^ and then centrifuged at 4 °C with an SW40 rotor for 12 h at 32,000 rpm. The profiles were detected through the continuous monitoring of the absorbance at 260 nm using a Biocomp Gradient Master/AKTA pure.

### 2.3. Cryo-EM Sample Preparation and Data Collection

Samples of 30S_1mM_, 30S_2.5mM_, and 30S_10mM_ with the corresponding peaks were collected and dialyzed to remove the sucrose. The fresh samples were diluted to 300 nM in a corresponding buffer, and then 2.5 μL of isolated particles was applied to glow-discharged R1.2/1.3 holey carbon grids with 2–4 nm continuous carbon film on top. After 30 s of waiting, the grids were blotted for 3 s and plunged into liquid ethane using a Vitrobot device (FEI) operating at 4 °C and 100% humidity.

Micrographs were collected on a Titan Krios G3i operating at 300 kV with a Gatan K3 Summit. Data acquisition was performed using the software EPU, with a nominal magnification of 81,000× *g*, which yields a final pixel size of 1.095 Å on the object scale (defocus ranging from –1.5 μm to –2.5 μm). For each micrograph stack, 30 frames were collected, for a total dose of 30 electrons per pixel. For the EDTA-treated sample, micrographs were recorded on a Titan Krios G3i operating at 300 kV with a Gatan K2 Summit.

### 2.4. Image Processing

Motion correction on the micrograph level was performed with MotionCorr2 [25]. The program CTFFIND4 was used to estimate the contrast transfer function parameters [26]. Image processing, including micrograph screening, particle picking, 2D and 3D classification, refinement, and postprocessing, were performed with RELION 3.1.0 [27]. 

For the 30S_1mM_ sample, a total of 497,673 particles were subject to a cascade of 2D and 3D classification. After one round of 3D classification, 440,474 particles were subjected to 3D auto-refine and then subjected to 3D classification with a mask on the decoding centers, S12 and h17, respectively, in which the alignment of the particles was omitted.

For the 30S_2.5mM_ and 30S_10mM_ samples, a total of 360,621 and 349,073 particles were subjected to several rounds of 2D/3D classification to remove the non-ribosomes and bad particles. Finally, 139,475 and 146,689 particles, respectively, were used for the 3D reconstructions. 

### 2.5. Model Building

A high-resolution cryo-EM structure of the *E. coli* ribosome (PDB:7k00) was used as the initial model, and rigid-body fitted to the density map using Chimera and Coot 0.8.9 [4,28,29]. The ribosomal protein and 16S rRNA were then fitted individually as rigid bodies and manually adjusted for the best fit between the map and the model. For the additional h17 density on the map, h17(437–497) was extracted from 7k00 23s rRNA and fitted as a rigid body. To illustrate the shifting of the rRNA helices, h6, h16, h18, h24, h27, and h45 were fitted to the maps using rigid-body fitting followed by real-space refinement in Coot [28].

### 2.6. CryoDRGN Analysis

To study the correlations of the structural distortion between different blocks, we exploited cryoDRGN’s powerful generative model to analyze the structural heterogeneity [30]. The particles were processed in RELION 3.1.0 until 3D refinement was achieved with the mask on the body, which contained 440,474 particles. Then, the results were applied for cryoDRGN training, in which the particles were downsampled to a box size of 256 (1.638 Å per pixel). The networks for the datasets were trained with an eight-dimensional latent variable.

For the subunit occupancy analysis, 500 volumes were sampled from the latent space. Then, the body domain of 30S was split into 32 blocks using a PDB file that was rigid-body-fitted to the refined map, including coordinates for two h17 helices (h17_in_ and h17_out_). The generated atomic models were used to create masks corresponding to each of the rRNA helices and ribosomal proteins. Then, these 32 masks were applied to each of the 500 volumes in turn, and finally, the occupancy of the density was calculated for each block, and the correlation between them was calculated by hierarchical clustering analysis.

## 3. Results

### 3.1. Ribosomal Subunits Are Destroyed by EDTA Treatment

It is known that metal ions are essential for stabilizing the structure of ribosomes and maintaining their activity. EDTA-treated ribosomes of *E. coli* have been reported to be Y- and X-shaped for the unfolded 30S and 50S [31], respectively. To symmetrically study the structures of ribosome subunits under the conditions of low Mg^2+^ concentrations, crude ribosomes from the *E. coli* strains BL21 and MRE600, as well as yeast and human cells, were analyzed by sucrose gradient sedimentation containing 10 mM EDTA (Figure 1a–d). Gradients with 10 or 2.5 mM Mg^2+^ were used as controls (gray curves in Figure 1a–d). The profiles of the gradients with EDTA showed peaks corresponding to the small and large subunits shifted towards the top of the gradient, which indicated a dramatic decrease in the molecular weight or particle size of the subunits. Then, the two shifted peaks of *E. coli* BL21 and MRE600 were collected and pooled (Figure 1a,b). CryoEM imaging showed that the subunits obtained from the 10 mM EDTA gradient were largely destroyed, with most, if not all, of the ribosomal proteins being dissociated and the rRNA being exposed in extended states (Figure 1e,f, white arrows). The shifting of the subunit peaks in the sucrose gradient profiles and the extended shapes of the rRNA observed by cryoEM imaging indicate that the extraction of Mg^2+^ from both prokaryotic and eukaryotic ribosomes by EDTA destroys the structure of ribosomes. Following this, the destruction process was further studied using 30S subunits as a model with a series of cryoEM structures. 

### 3.2. Structural Distortion of 30S at a Low Magnesium Ion Concentration

Magnesium ions are the most abundant ions present in ribosomes [1]. To further study the distortion process of the structure of ribosome subunits, relevant fractions in a sucrose gradient sedimentation of the *E. coli* strain were collected and subjected to cryoEM analysis (Figure 2a). In this study, we isolated the 30S particles from a reference-free 2D classification strategy in Relion 3.1.0 for intensive study (Figure 2d). Typical images of the 30S_1mM_ peak showed 30S profiles with a flexible h17 helix and smear tracks of the 30S head (Figure 2d, red arrows), showing that the low Mg^2+^ concentration affected the 30S structure. To further confirm this hypothesis, we collected the 30S peak from sucrose gradients with 2.5 and 10 mM Mg^2+^, respectively, for cryoEM analysis (Figure 2b,c). The 2D averages for 30S_2.5mM_ showed a better head and h17 helix (Figure 2d, 2nd line), and the 30S_10mM_ showed a far more stable 30S head and a fold-in h17 helix (Figure 2d). These observations indicate the important role of Mg^2+^ ions in maintaining the structure of rRNA in the 30S subunit. Particles from these three datasets of 30S_1mM_, 30S_2.5mM_, and 30S_10mM_ were then extracted for further analysis (Appendix A Appendix A).

### 3.3. Overall Structures of the 30S Subunits under Different Mg^2+^ Concentrations

As seen above, the 30S at 1 mM Mg^2+^ showed a flexible h17 and a blurry head. To explore the structural distortion in more detail, the particles were subjected to a 3D reconstruction using Relion 3.1.0 [27], with a mask on the 30S body to eliminate the interference of the blurry head-on alignment, resulting in a set of structures with a resolution of 3–5 Å (Figure 2e and Appendix A Appendix A). Consistent with the 2D averages seen in Figure 2d, 30S in the 1 mM Mg^2+^ condition showed the same conformation, with a poorly aligned head and well-defined body (Figure 2e). Spahn et al. proposed that the rotation of the head of the small subunit directs the movement of the tRNAs to the P and E sites [32]. Hence, the head of 30S plays a vital role in translation, and a comparative structural analysis of 55 ribosome structures showed that the 30S head rotated by 0–21 degrees related to the body part, as determined by the E-R method [33,34]. Using multi-body refinement in Relion 3.1.0, a program used to classify heterogeneous cryo-EM structures, the 30S from 1 mM Mg^2+^ showed a much broader range of rotation in its head (Appendix A). Even with a mask on the head, the reconstructions still failed to show a clear map of the rRNA or proteins (Appendix A), which means that low Mg^2+^ conditions can cause the 30S head to become even more flexible on its own accord. Meanwhile, compared to the 30S at 2.5 and 10 mM Mg^2+^ concentrations (Appendix A), the first thing that we noticed in the 1 mM Mg^2+^ reconstructions was the absence of h44 (Figure 2e), which has almost disappeared in 2.5 mM Mg^2+^ as well (Appendix A). Flexible regions, including the decoding center, h16, h17, and platform, and a weak S12 density were also observed on the map (Figure 2e). These functional regions mature at different timepoints when 30S assemble, requiring plenty of Mg^2+^ to stabilize their positions [35].

### 3.4. Movement of Incompact Helices and Loss of S12 in the Decoding Center

In 30S, the decoding center, which is composed of h27, h28, h1, h2, the upper part of h44, and h45, offers a place for interaction between mRNA and tRNA, contributing to the fidelity of decoding through the monitoring of codon-anticodon base pairing [20,36]. To separate the different conformations contained in the 30S reconstructions of 1 mM Mg^2+^, we first performed non-alignment 3D classification based on the auto-refined angles with a mask on the decoding center, resulting in different reconstructions with diverse structural distortions in the decoding center. Here, the maps are named as h27_in-1, 2,_ and h27_out-1–4_ according to their h27 positions, and three main classes are obtained (Figure 3a–c and Appendix A Appendix A). Approximately 7% of particles (h27_out-3_) showed long-distance (~26 Å) shifting towards h18, and, accordingly, h18, h24, and h45 became more flexible (Figure 3a). The h27_out-4_, with 15% particles, showed short-distance (~19 Å) shifting compared to h27_out-3_ (Figure 3b), whereas the h17_in-2_, which contained approximately 56% of the total particles, showed a stable h27 that was fixed to the mature state (Figure 3c). The other three states showed a different level of movement, according to which h27_out-1_ and h27_out-2_ were similar to h27_out-4_, containing a short-distance-shifted h27 helix (Appendix A Appendix A), and the h27_in-1_ state had a near-mature h27 helix, as seen in h17_in-2_ (Appendix A Appendix A). 

In the states separated by the mask of the decoding center, in addition to the movement of h27 and the correlated swing of h18, h24, and h45, we also observed a weakened density in the S12 protein. S12 is located near the decoding center. It is composed of two distinct parts, including the N-terminal extension and the conserved C-terminal globular region [37,38]. S12 plays a pivotal role in decoding functions and is a key mediator in maintaining the fidelity of translation on the ribosome. Research has shown that S12 is important for the inspection of codon–anticodon pairings at the ribosomal A site [39]. The N-terminus of the protein binds the solvent surface of the SSU, with the extension in contact with the rRNA dense regions, ending with a C-terminal globular region localized at the inter-subunit face of the SSU, which means that S12 plays a vital role in maintaining the small subunit structure [40]. To further understand the binding of S12 in the decoding center, we then exploited 3D classification using a mask on the S12 protein alone, resulting in 10 different classes showing various occupancy levels of S12 (Figure 3d,e and Appendix A Appendix A). In the states of S12_4_ and S12_9_, which contained approximately 14% of the total particles, the S12 protein was completely missing (Figure 3d and Appendix A Appendix A). Along with the disappearance of the S12 protein, helix h18 and h27 in these two states moved by approximately 7 and 18 Å, respectively (Figure 3d). State S12_10_ had 53% of particles and showed a high level of occupancy at the S12 density, with slight shifting of h18 and h27 (Figure 3e). States S12_1–8_, except S12_4_ and S12_6_, also had a partial density of S12, which means that low Mg^2+^ only partially destabilized the interactions between S12 and the rRNA (Appendix A Appendix A). State S12_6_ had no density in the position of S12 but showed extra density above S12, probably being in an intermediate state (Appendix A Appendix A). We should also note that, due to the steric hindrance, the helix h16 lying close to h18 moved further when S12 was missing compared to the states with S12 proteins (Figure 3d,e). 

### 3.5. Movement of h17 towards h6

In addition to the decoding center, we could also observe an extra density around h17 toward h16 (Figure 2d). Applying a mask to h17 alone, we identified four major classes from the particles of 30S at 1 mM Mg^2+^ (Appendix A). The map named h17_in_ is identical to the mature 30S, in which h17 interacts with h10 through its tip from A465 to C470 and with h15 through the central part of h17. Interactions between the central part of h17 and the positively charged residues of the S16 protein were also observed (Figure 4a,b). The second one, containing 38% of the particles and named h17_out_, had a clear helix-shaped density that moved outwards by approximately 54 degrees, in which the relocated h17 could be fitted, folding as a bridge that connected the tip of h6 and junction of h16/h17 (Figure 4c). In this conformation, the tip of h17 (G462–U464) is close to the helix h6 (U85–G86) and stabilized by the interactions between them (Figure 4d). Since h16 and h17 are connected and form a long helix in the 30S subunit, we also compared the h16 helix in these two states (Appendix A Appendix A). A bent h16/h17, as one can observe in a mature 30S, was observed in the h17_in_ state, and a long, straighter helix was found in the h17_out_ state. Whether h17 is located within or shifted out, h16 swayed at the same angle in both states, which means that the junction of h16 and h17 is very flexible; thus, the movement of h16 is independent of h17 shifting. The third class, called h17_in-2_, showed a blurry density between the h17_in_ and h17_out_ conformations, being closer to h17_in_, meaning that it should be in the intermediate state (Appendix A Appendix A). The remaining 23% of particles comprised a 30S subunit with an invisible h17 and most of its helices, probably due to its highly flexible rRNAs.

### 3.6. Correlation of the Structural Distortions between Different Blocks

Except for the decoding center and h17, as mentioned above, some other parts of the 30S also became less stable under a low Mg^2+^ concentration. To determine whether there are correlations between the destabilization of these rRNA helices and ribosomal proteins, we exploited cryoDRGN’s powerful generative model by sampling 500 volumes from the latent space for a total of 440,474 particles with 35 learning epochs [30] (Figure 5a). In this strategy, the coordinates of the 30S head were first removed since no clear density could be assigned to it. Then, the body was split into 32 blocks using a PDB file that was rigid-body-fit to the refined map, including coordinates for two h17 helices (h17_in_ and h17_out_). Each structural block contained an individual rRNA helix, or ribosomal protein. The occupancy of the density was calculated for each block, and the correlation between them was calculated by hierarchical clustering analysis (Figure 5a). Here, we observed that h44 had totally disappeared, and the occupancy of h17, S11, and S21 was significantly decreased in most of the volumes. Additionally, approximately half of the 500 volumes had a poor overall density, revealing the global flexibility of 30S at 1 mM Mg^2+^. For the other half of the volumes that had a higher occupancy for most of the blocks in the 30S body, we could identify correlations between the unstable helices and proteins (Figure 5a,b). Hierarchical clustering showed that proteins S11 and S21 were omitted simultaneously in most cases, and h23, h24, and h45 showed a similar pattern. It is worth noting that the occupancies of h27 and protein S12 in the decoding center had different distributions in the 500 volumes compared to h17 (Figure 5a). We further analyzed the distributions of all the particles in the states identified using different masks (Figure 5c,d). The particle number in each state was calculated and analyzed by the chi-squared test. This showed that when h27 is in states h27_out-1,2,3_ or h27_in-1_ and S12 is in states S12_1–8_, the h17 helix tends to be flexible, whereas in the h17_out_ particles, the h27 tends to be fixed in the mature position (h27_in-2_) and S12 is complete (S12_10_). On the other hand, h27 tends to shift outward, and S12 shifts away in the two h17_in_ states. These results showed a negative correlation between the movements of h17 and h27/S12, similar to the hierarchical clustering results (Figure 5a).

### 3.7. Mg^2+^ Is Essential for Ribosome Structure Stability

To further investigate the effect of the Mg^2+^ concentration on ribosome assembly, we performed sucrose gradient analysis with different levels of Mg^2+^ (Figure 6). As observed at 1 mM Mg^2+^ (Figure 2a), the 0.5 mM Mg^2+^ condition also completely separated the 50S and 30S subunits (Figure 6a), whereas at the 2.5 and 5 mM Mg^2+^ concentrations, the 70S ribosome formed gradually (Figure 2b and Figure 6b). Our structural study of the 30S peak at 2.5 mM Mg^2+^ also showed a barely visible h44 and a flexible h17 (Appendix A). With 10 mM Mg^2+^ or above, the 70S became very stable (Figure 2c and Figure 6c). To determine whether the 30S and 50S purified from 1 mM Mg^2+^ were still functional, we collected and incubated these two peaks, followed by reloading the gradient with 10 mM Mg^2+^. The results showed that most of the 30S and 50S could form an intact 70S (Figure 6d). The remaining 30S that could not interact with 50S might be the result of S12 protein dissociation, as observed in our structures (Figure 3). Analysis of the ribosomal proteins in the 30S peak by SDS-PAGE showed several weakened bands at 0.5 and 1 mM Mg^2+^ compared to 10 mM Mg^2+^ (Appendix A Appendix A). These could be the results of weak binding to the rRNA at a low Mg^2+^ concentration, followed by a falloff from 30S when the concentration by centrifugation was applied. 

## 4. Discussion and Conclusions

The data presented here, ranging from 1 mM to 10 mM Mg^2+^, represent a model of the process of *E. coli* 30S unfolding during Mg^2+^ decrease (Figure 7). Initially, the 30S structure is completely intact at 10 mM Mg^2+^. When the concentration of magnesium ions decreases, the rotation angle of the head increases accordingly, along with the loosening of the proteins and RNA helices. Here, h44 is most sensitive to magnesium ions, first becoming unstable, and then h17 begins to swing away (Figure 7, dashed box at 2.5 mM Mg^2+^). With the Mg^2+^ decreasing to 1 mM, the rRNA helices h16, h18, and h27 in the decoding center also become unstable, and protein S12 starts to dissociate from the 30S subunit (Figure 7, dashed boxes in 1 mM Mg^2+^). When there are no magnesium ions at all, that is, after EDTA treatment, the 23S rRNA becomes completely unfolded, with a loss of ribosomal proteins. Most likely, the secondary structures are still stable, except for those helices with only a few canonical Watson–Crick base pairs [8]. Our cryoEM structures showed important tertiary interactions stabilizing the rRNA fragments formed/destroyed at different Mg^2+^ concentrations, reflecting differences in stability between different regions. This is consistent with the calculation showing that the midpoint, the Mg^2+^ concentration at which individual secondary or tertiary interactions occur, is not unique, and the coordination of Mg^2+^ with rRNA is nucleotide specific and does not occur in a random, diffusive manner [8].

The magnesium ion, which is the most abundant of the divalent cations in living cells, has an irreplaceable function in stabilizing the secondary structure of ribosomal RNA, binding ribosomal proteins to the ribosome, and ribosomal interaction due to its high charge density and relatively small ionic radius (0.6 Å) [3,41,42]. It is known that the removal of Mg^2+^ from ribosomes results in a non-functional ribosome, especially in the case of EDTA-treated *E. coli* ribosomes, which show a different structural assembly manner [2]. In this study, we used cryoEM to study the disassembly process of the small subunit of the *E. coli* ribosome and showed a series of snapshots of the distortional 30S, which revealed a sequential movement of the rRNA helices and ribosomal proteins in the 30S subunits. Several cryoEM structures suggest that the pre30S/pre40S particles transit through a vibrating state [43,44,45]. We concluded that the movements observed in this study using a low Mg^2+^ concentration are correlated with the structural rearrangements observed during 30S/40S maturation.

Firstly, all the reconstructions obtained in this study lacked a clear h44, which means that this helix is totally moved out at a 1 mM Mg^2+^ concentration, although we could observe weak densities above the decoding center in some of the states responsible for the moved h44. The helix h44 is directly involved in mRNA decoding, as is the formation of two inter-subunit bridges (B2a and B3) that participate in association with the large ribosomal subunit [46,47]. In the process of the decoding center’s maturation, h44 forms before the correct base pairing of h28 and the linker h28/h44. The assembly factors RimP and RsmA then lift h44 to access h28 for its refolding with the help of RbfA. In the last step, the decoding center, with an h44 in the final position, is checked by RsgA, which means that the placing of the h44 helix is the last step in 30S maturation. Thus, it makes sense that h44 is the first part to be dissociated when the 30S structure becomes unstable [45]. In the decoding center, we also observed a movement of h27 by approximately 26 degrees towards h18 due to the disappearance of h44. Helix h27, termed the switch helix, is packed groove-to-groove with the upper end of h44, which is the target of amino-glycoside antibiotics [20,48]. A recent study showed that h27 and h21 had larger fluctuations than other helices in the central domain of 16S rRNA due to their absence in extensive tertiary interactions [8]. In other words, h27, under conditions of low Mg^2+^ concentration, also becomes more flexible than other helices. In addition to h27 and h44, the surrounding helices h18, h24, and h45 also showed significant movement under the 1 mM Mg^2+^ conditions.

Secondly, in part of the reconstruction, the ribosomal protein S12 dissociated from the decoding center. S12 is the third binding r-protein according to the Nomura assembly map [49,50]. The C-terminal globular region of S12 is close to the decoding center, and it is unique among SSU r-proteins, because S12 is the only protein located on the RNA-rich surface that interacts with the large subunit [20]. It is clear that the ribosomal protein S12 plays a pivotal role in tRNA selection by the ribosome [39,51]. Crystal structures have revealed that the closed conformation of the 30S subunit is stabilized by interactions between the conserved amino acid residues of ribosomal protein S12 and 16S rRNA h44 at the decoding site [52]. Recent simulation studies showed that the central domain of the 16S rRNA without ribosomal proteins unfolds at low Mg^2+^ concentrations of roughly 2 mM.

Thirdly, an approximately 54 degree rotation was observed in h17, which generated novel interactions between the tip of h17 and h6. In the mature 30S, h16/h17 comprises one of the three long helices in the 30S subunit and interacts with h18/h15, which is situated beside it and forms the backbone of the entire body. The three major longitudinal elements, h44, h16/h17, and h7, act as structural pillars that extend over 110 Å, and not only stabilize the body but also transmit conformational changes and displacements over a very long distance [20]. Here, we observed that h17 started to shift towards the tip of h6, even at 2.5 mM Mg^2+^, whereas this movement did not affect the connected h16. Furthermore, the 30S head and related helices in the neck region were too flexible to be defined in our reconstructions. In the translation process, large-scale rotation of the head domain is required for the movement of mRNA and tRNA translocation [33,34]. 

In *E. coli*, the Mg^2+^ concentration is reported to be approximately 1–5 mM, whereas the total Mg^2+^ concentration, including the Mg^2+^ chelated by biological molecules, is around 100 mM [16,53]. Recently, the ribosome itself was shown to be involved in Mg^2+^ homeostasis [5,54]. A high-resolution crystal structure of the 70S showed that it contains more than 170 Mg^2+^ ions bound tightly to the ribosome [40]. Additionally, a 2.0 Å cryoEM structure of the *E. coli* ribosome identified 309 Mg^2+^ in 70S with 93 Mg^2+^ in 30S [4]. Counting the Mg^2+^ associated loosely or through out-sphere interactions showed that more than 585 Mg^2+^ are predicted to bind the elongation complex of the ribosome [55]. Studies of *B. subtilis* showed that the total cellular Mg^2+^ concentration decreased in proportion to the amount of 70S ribosome when *B. subtilis* lacked an individual copy of the rRNA operons [12,15]. Combined with our observation of the sequential structure distortions of 30S rRNA in this study, a corollary is that the ribosome is a reservoir of Mg^2+^, and when the cellular free Mg^2+^ decreases, the ribosome first releases the Mg^2+^ on the surface, which only involves a small portion of rRNA helices, such as h44 and h17 in the 30S, and this results in a reversible ribosome. Meanwhile, the mechanisms through which ribosomes actively participate in Mg^2+^ homeostasis should be elucidated in detail in future studies. 

It has been reported that the most ancient parts of the ribosome are the PTC and the ribonucleotides, which depend heavily on metal ions for their structural stability [56,57]. Moreover, the twofold pseudo-symmetry in and around the PTC, which is composed of RNA and Mg^2+^, has been suggested to be the structural origin of the ribosome [57,58,59,60]. Mg^2+^ is also essential for the small subunit functional regions. According to our structures, the *E. coli* 30S and 50S subunits separate and subsequently unfold when the Mg^2+^ concentration is below 1 mM. Ribosomes are usually purified at 10 mM Mg^2+^ to ensure that they are close to their natural state, but subunit separation occurs at 1 mM Mg^2+^ [61,62]. One should pay attention to the fact that protein S12 starts to dissociate from 30S under this condition, and long-time incubation at 1 mM Mg^2+^ should be avoided. These results can provide guidance for studies of ribosome structural and functional stability as well as the process of ribosome assembly.

## Figures and Tables

**Figure 1 biomolecules-13-00566-f001:**
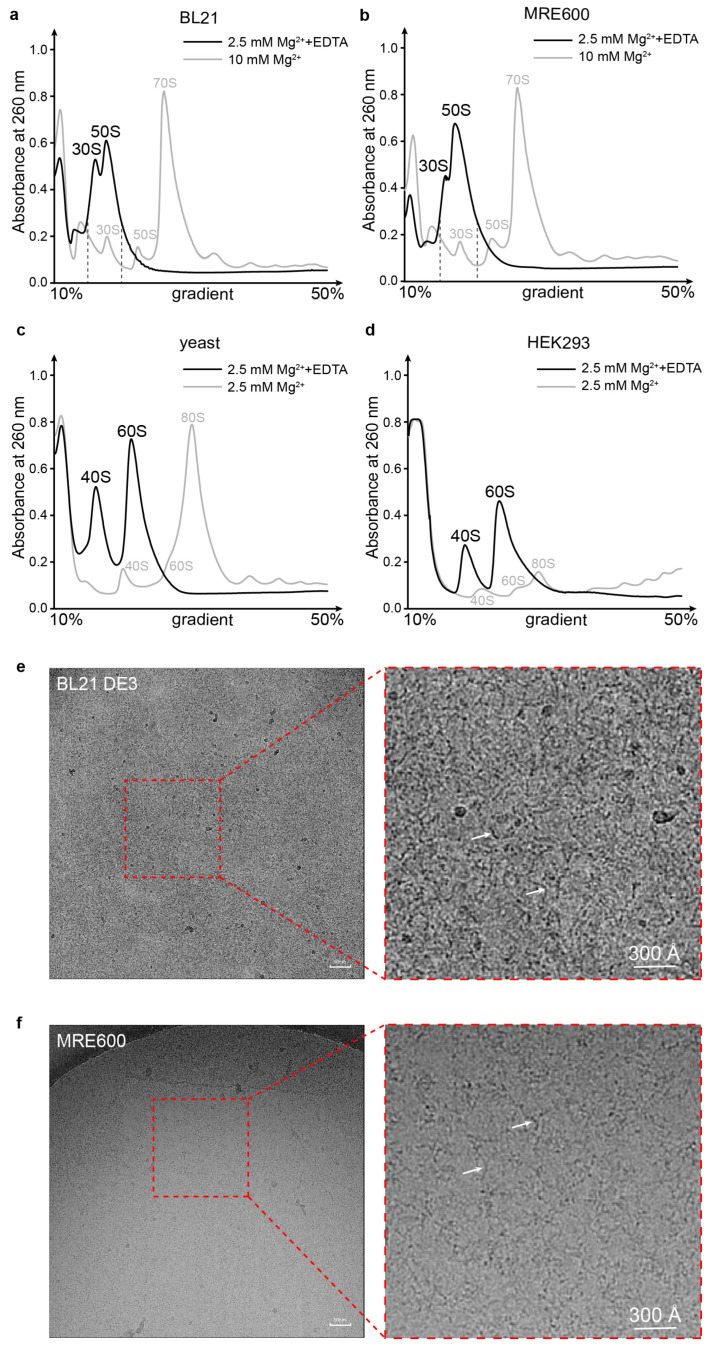
Subunits of the 70S ribosome were destroyed by EDTA. (**a**,**b**) *E. coli* cells of the BL21 DE3 (**a**) and MRE600 (**b**) strains were collected at OD_600_ = 0.6 and further analyzed with a sucrose gradient in conditions with or without 10 mM EDTA. Shifting of the peaks was indicated, and the 30S and 50S peaks from the gradient with additional EDTA were collected (dashed line labeled) for cryoEM analysis. (**c**,**d**) Yeast BY4742 cells (**c**) and HEK293F cells (**d**) grown to exponential phase were collected and disrupted by a French press and homogenizer, respectively. Cell extracts were then analyzed with a sucrose gradient under conditions of 2.5 mM Mg^2+^ or 10 mM EDTA. (**e**,**f**) CryoEM images representing the typical particle shapes (white arrows) for BL21 DE3 and MRE600 (**f**) strains. The scale bar is labeled in white.

**Figure 2 biomolecules-13-00566-f002:**
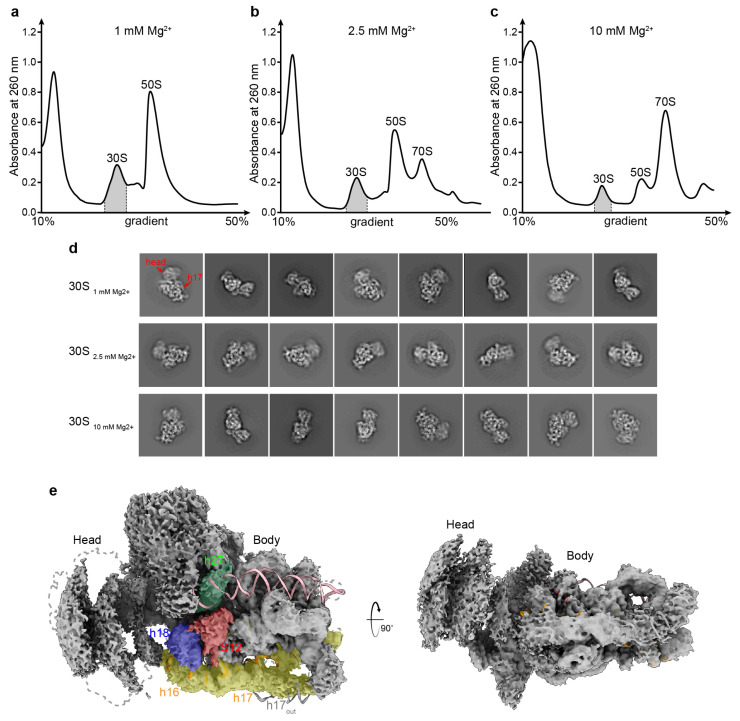
A 30S subunit at 1 mM Mg^2+^ has a flexible h44, h17, decoding center, and head. (**a**–**c**) Sucrose gradient sedimentation profile of *E. coli* ribosomes under 1 mM (**a**), 2.5 mM (**b**), and 10 mM (**c**) Mg^2+^ conditions. The 30S peak indicated in gray shadow was collected separately for cryoEM analysis. (**d**) Reference-free 2D classification averages for 30S particles under different conditions, as shown in (**a**–**c**). The flexible h17 and head are labeled with red arrows. (**e**) The overall structure and map of a 30S subunit at 1 mM Mg^2+^ concentration are represented in cartoon and surface, respectively. The rRNA helices h16/h17, h18, h27, and h44 and the S12 protein are colored in yellow, blue, green, magenta, and red. The dashed line represents the mature 30S under 10 mM Mg^2+^ conditions in this study.

**Figure 3 biomolecules-13-00566-f003:**
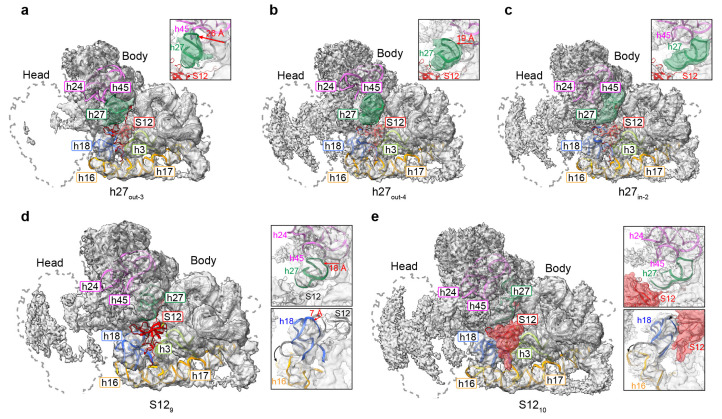
A low Mg^2+^ concentration destabilizes the decoding center and causes a loss of S12 protein. (**a**–**c**) Three representative reconstructions were classified from 30S particles at 1 mM Mg^2+^ by applying a mask to the decoding center. A completely out-shifted h27 ((**a**), h27_out-3_), a partially out-shifted h27 ((**b**), h27_out-4_), and h27 in its original position ((**c**), h27_in-2_), are represented in cartoon and surface (green) for their rigid-body-fitted structure and density map, respectively. (**d**,**e**) Two representative reconstructions with missing (**d**) and fully occupied S12 protein ((**e**), red), were classified from 30S particles at 1 mM Mg^2+^ by applying a mask to S12, as shown in the cartoon and on the surface. The rRNA helices h3 (limon), h16/h17 (yellow), h18 (blue), and h24/h45 (magenta) are also labeled according to the positions of the decoding center. Details of the interactions and movement (black arrows) of h27 and S12 are represented as inserted subfigures. A dashed line represents the mature 30S under 10 mM Mg^2+^ conditions in this study.

**Figure 4 biomolecules-13-00566-f004:**
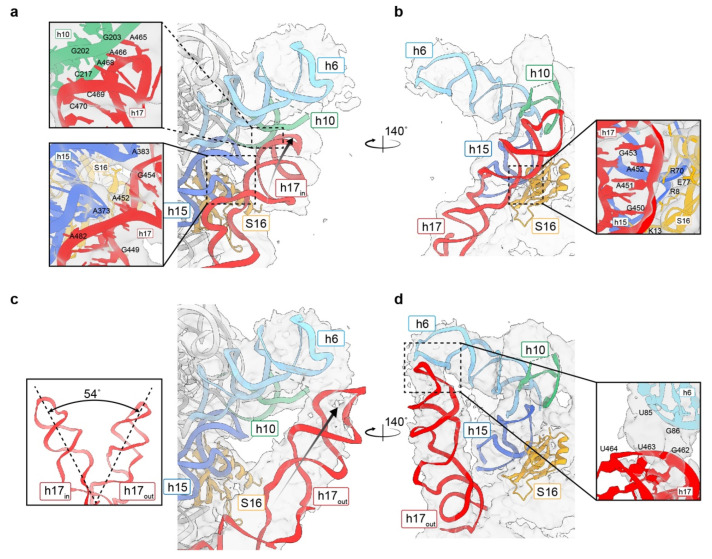
The h17 helix shifted outwards to the tip of h6. (**a**) Interactions between h17 (red), h10 (green), and h15 (blue) in the reconstruction of h17_in_. The direction of h17 is labeled with a black arrow. (**b**) Interactions between h17 and S16 proteins in the reconstruction of h17_in_. (**c**) Shifting of h17 from h17_in_ to h17_out_. The shifted angle was measured in Chimerax, and the direction of h17 in the h17_out_ reconstruction is labeled with a black arrow. (**d**) The interactions between the out-shifted h17 and the tip of h6.

**Figure 5 biomolecules-13-00566-f005:**
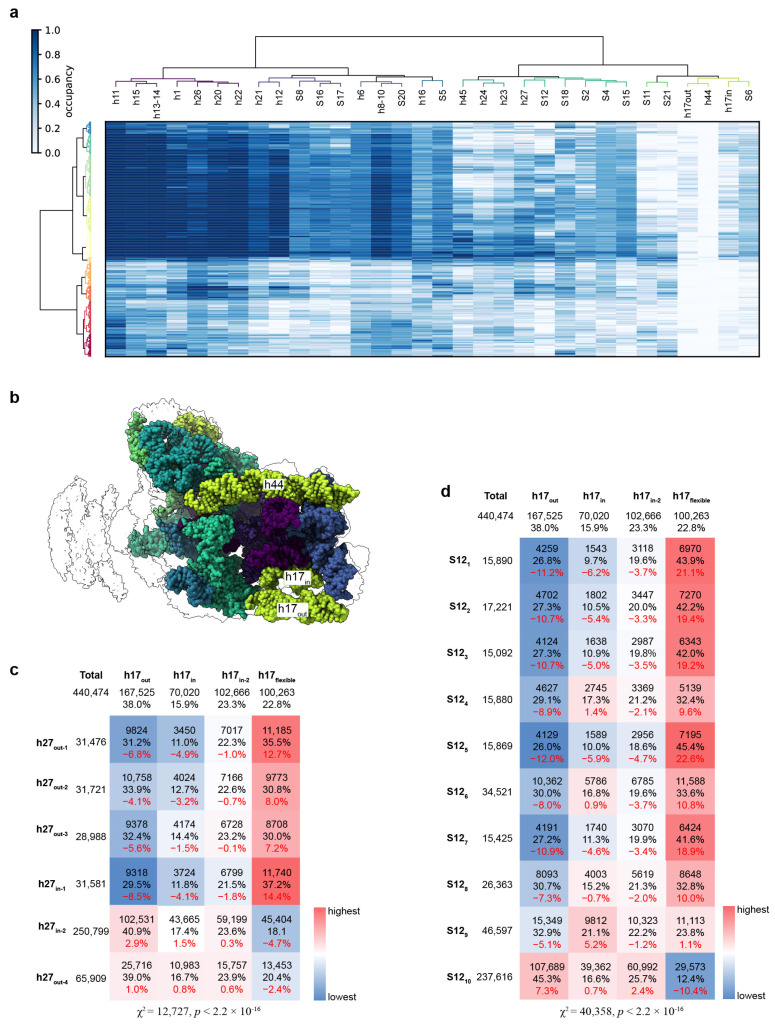
The movements of the different structural blocks are correlated. (**a**) Occupancy analysis of 30S_1mM_ particles displayed as a heatmap. Rows (500) correspond to sampled density maps, and columns (32) correspond to structural elements defined by the atomic model. Only the 30S body was analyzed. (**b**) Atomic models of the 30S subunit used for subunit occupancy analysis are colored according to the structural blocks defined through hierarchical clustering in (a). Structural features of interest are annotated. (**c**) Correlation between h17 and h27 in the reconstructions classified with different masks. The mask on h17 defined four different states of h17, and the mask on the decoding center defined six states of h27. Particles in each state were selected, and the distribution of particles in different h17 conformations was calculated. For each h27 state, the difference in distribution compared to the total particles is colored in red. (**d**) Same as (**c**), but the correlation between h17 and S12 was calculated.

**Figure 6 biomolecules-13-00566-f006:**
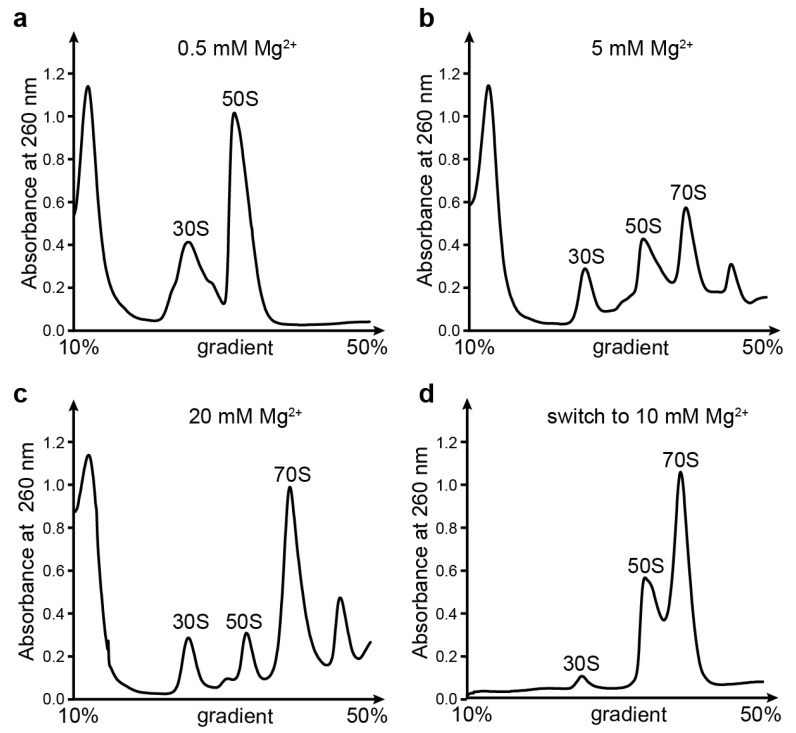
The effect of Mg^2+^ concentration on the assembly of 70S. (**a**–**d**) Sucrose gradient profiles for crude 70S at 0.5 mM (**a**), 5 mM (**b**), and 20 mM (**c**) Mg^2+^ concentrations. (**d**) Profile of 30S and 50S purified from 1 mM Mg^2+^ in a gradient containing 10 mM Mg^2+^. Fractions corresponding to 30S and 50S under 1 mM Mg^2+^ conditions (Figure 2a) were pooled, and sucrose was removed, followed by reloading onto a sucrose gradient containing 10 mM Mg^2+^.

**Figure 7 biomolecules-13-00566-f007:**
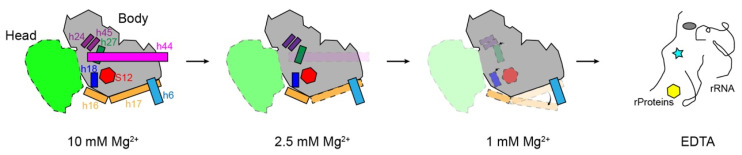
A model of 30S structural distortion at a low Mg^2+^ concentration. Here, 2.5 mM Mg^2+^ first causes the destabilization of h44 and partial loosening of h17, and the 30S head becomes more flexible. A further decrease in Mg^2+^ to 1 mM shifts h17 toward the tip of h6, and h16, h18, and h27 become flexible at the same time. Ribosomal S12 near the decoding center starts to leave, causing an irreversible 30S if no additional protein is supplied. EDTA incubation was used to extract all the Mg^2+^ from ribosomes and completely destroy the 30S subunit, release the ribosomal proteins, and linearize the 16S rRNA.

## Data Availability

Electron microscopy maps were deposited in the Electron Microscopy Data Bank under accession codes EMD-34987, EMD-34985, and EMD-34986 for 30S at 1, 2.5, and 10 mM Mg^2+^, respectively, with EMD-34988 for h17_flexible_, EMD-34989 for h17_out_, EMD-34990 for h17_in-2_, and EMD-34991 for h17_in_.

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
