# Peer review of "Structural Insights into the Distortion of the Ribosomal Small Subunit at Different Magnesium Concentrations"

_biomolecules, 2023, doi:10.3390/biom13030566_

Round 1

Reviewer 1 Report

In this manuscript, Ting Yu and colleagues have used cryo-EM and single particle analysis to monitor the de-structuration of prokaryotic small ribosomal subunits (30S subunits) caused by low Mg2+ concentrations. They show how lack of this divalent cation distorts rRNA helix h44, the decoding center, rRNA helix h17 as well as the 30S head and platform domains. They analyze the sequence with which the structure of ribosomal 30S subunit structure is disassembled upon low Mg2+ concentration.  They also provide various evidences that reveal how, depending on Mg2+ concentration, distorted 30S subunits can either recover their 3D structure and be integrated into 70S ribosomes, or lead to an almost completely unfolded 16S rRNA, probably devoid of ribosomal proteins. Overall, this work is solid, well designed, and "visually" proves (by using cryo-EM investigations) the capital role played by Mg2+ and other divalent cations in the structuration of small ribosomal subunits, that will allow its correct functioning during translation.

In order to strengthen the clarity and the impact of this study, I would nonetheless like to raise the following comments/remarks:

-a very general comment, there seem to be a lot of minor language errors as well as incorrect turns of phrase, this should be carefully checked to increase the clarity of the manuscript.

-p. 4, results paragraph “30S and 50S of E. coli ribosomal subunits are destroyed by EDTA treatment”: why do the authors introduce the function of helicase DbpA here? It might be more relevant to talk about it on p.5 (paragraph “Structural distortion of 30S at low magnesium ion concentration”), when examining the effects of low Mg2+ on E. coli strains expressing DbpA mutants with or without rescue. Otherwise, the message delivered in this first paragraph is less clear.

-p.4, same paragraph: figure 1 c and d: cryo-EM images show no structured ribosomal subunits, but harbor elongated shapes that could correspond partially unfolded rRNA. However, how can the authors affirm that ribosomal proteins were extracted from the unfolded rRNA? This sentence should be modified. Moreover, this sentence could mistakably lead readers to believe that authors performed protein and/or rRNA extraction from their sucrose gradient before putting rRNA on cryo-EM grids.

-p.4, final sentence and Figure S1: could the authors clearly state their conclusions when performing similar experiments on eukaryotic cells? For instance, do the authors think that lack of Mg2+ and divalent cations also distort 3D structures of eukaryotic ribosomal subunits?

-p. 5/6, results paragraph “Structural distortion of 30S at low magnesium ion concentration”

Here again, justification of the use of a mutant version of DbpA to study the effect of magnesium depletion on 30S subunits is difficult to grasp. Why using this mutant if it is not involved in 30S subunit biogenesis? Furthermore, Fig. 2 displays gradient values that are different between fig 2a (20-40%) and panels 2b and 2c (10-50%); it is thus complicated to understand the correspondence between peaks. In figures 2a and 2b, how do the authors attribute the peaks to 30S, 45S and 50S? If DbpA is involved in 50S biogenesis, could the so-called 45S peak correspond to a diminished number of 50S subunits, while the so-called 50S peak actually be a full 70S ribosome? These remarks do not affect the strength of structural analysis presented in figure 2d and 2e. Could the authors clarify which of the “particles from the three datasets were extracted for further analysis”? Do they mean the three datasets that had were obtained with a concentration of 1mM Mg2+?

-p13, discussion chapter. The authors propose a very plausible model of 30S distortion caused by low Mg2+ concentration. How does this model relate to published cryo-EM studies of in vitro assembly of prokaryotic ribosomal subunits? Furthermore, many cryo-EM structures of eukaryotic pre-ribosomes show structural rearrangements of the forming pre-40S particle, with huge movements of rRNA helix h44 and head domains during nucleolar assembly steps (cf for instance the review from Barandun and Klinge, Curr Opin Struct Biol 2018, or scientific articles such as Kornprobst et al, Cell 2016), as well as cytoplasmic ones (see Heuer et al., eLife 2017, Scaiola et al, EMBO 2018, Shayan et al, Molecules 2020). Can the movements herein observed using low Mg2+ concentration be correlated with the structural rearrangements observed during 40S assembly?

Author Response

Thanks for the three reviewer’s suggestions, changes have been made in the revised manuscript, mainly including:

  1. The statement of DbpA was moved from the Results to the Materias and Methods part.
  2. Methods for growing yeast and human cells were added.
  3. Details of data processing for 2.5 mM and 10 mM Mg2+ particles were added.
  4. Figures S1 was merged with Figure 1 and the description and discussion were added to the first paragraph of the Results.
  5. DbpA related images were removed from Figure 2. Instead, the 2.5 mM Mg2+ data was merged from orignal Figure 6.
  6. Figure S9 (30S5mM) was merged with Figure S4 (30S10mM) and the data processing procedures, FSC, local resolutions were added.
  7. SDS-PAGE analysis was performed in Figure S8 and discussed in the last paragraph of Results.

Answers for the reviewers’ comments were listed below.

Reviewer #1

In this manuscript, Ting Yu and colleagues have used cryo-EM and single particle analysis to monitor the de-structuration of prokaryotic small ribosomal subunits (30S subunits) caused by low Mg2+ concentrations. They show how lack of this divalent cation distorts rRNA helix h44, the decoding center, rRNA helix h17 as well as the 30S head and platform domains. They analyze the sequence with which the structure of ribosomal 30S subunit structure is disassembled upon low Mg2+ concentration. They also provide various evidences that reveal how, depending on Mg2+ concentration, distorted 30S subunits can either recover their 3D structure and be integrated into 70S ribosomes, or lead to an almost completely unfolded 16S rRNA, probably devoid of ribosomal proteins. Overall, this work is solid, well designed, and "visually" proves (by using cryo-EM investigations) the capital role played by Mg2+ and other divalent cations in the structuration of small ribosomal subunits, that will allow its correct functioning during translation.

In order to strengthen the clarity and the impact of this study, I would nonetheless like to raise the following comments/remarks:

-a very general comment, there seem to be a lot of minor language errors as well as incorrect turns of phrase, this should be carefully checked to increase the clarity of the manuscript.

Response: Thanks for the reviewer’s comment. We have sent the manuscript to MDPI author services (https://www.mdpi.com/authors/english) for high-quality editing. Certificate (Invoice ID: english-62756) can be supplied if required.

-p. 4, results paragraph “30S and 50S of E. coli ribosomal subunits are destroyed by EDTA treatment”: why do the authors introduce the function of helicase DbpA here? It might be more relevant to talk about it on

Response: In our study, the initial 30S particles were isolated from a ∆dbpA E.coli strain with DbpA(R331A) overexpressed from another project in our lab. We were trying to describe the fact that the 30S isolated from DbpA(R331A) strain is the same as the one from wild type BL21 strain, so that it can be used for 30S structure studies. To eliminate the misleading of DbpA, we have moved these sentences to the method part in the revised manuscript as below:

“In this study, 30S particles was purified from an E.coli strain with the mutant DbpA protein overexpressed, obtained in another project conducted by our research group. Since the structural distortion observed in these 30SDbpA under 1 mM Mg2+ was shown to be exactly the same as that of 30S purified from the wild-type BL21 strain (data not shown), the cryoEM data of 30SDbpA were used for reconstruction and designated as 30S1mM in this study. Wild-type E.coli BL21 and MRE600 were used for sucrose gradients and purification of 30S at 2.5 and 10 mM Mg2+.”

p.5 (paragraph “Structural distortion of 30S at low magnesium ion concentration”), when examining the effects of low Mg2+ on E. coli strains expressing DbpA mutants with or without rescue. Otherwise, the message delivered in this first paragraph is less clear.

Response: As for the comment above, sentences about DbpA was removed from Results part and reorgnized in Materials and Methods part.

-p.4, same paragraph: figure 1 c and d: cryo-EM images show no structured ribosomal subunits, but harbor elongated shapes that could correspond partially unfolded rRNA. However, how can the authors affirm that ribosomal proteins were extracted from the unfolded rRNA? This sentence should be modified. Moreover, this sentence could mistakably lead readers to believe that authors performed protein and/or rRNA extraction from their sucrose gradient before putting rRNA on cryo-EM grids.

Response: Thanks for the reviewer’s comment. Sentence has been rewritten to “CryoEM imaging showed that the subunits obtained from the 10 mM EDTA gradient were largely destroyed, with most, if not all, of the ribosomal proteins being dissociated and the rRNA being exposed in extended states”. We have conduted SDS-PAGE for the shifted peaks of gradient with 10 mM EDTA. Unfortunately, it was hard to separate these peaks with the free peak on the top of the gradient. Resulting lots of protein bands that was not belong to 30S/50S subunits (data was not shown). For the 30S peaks at 0.5, 1, and 10 mM, SDS-PAGE showed some weakened bands at low magnesium concentraion (Figure S9).

-p.4, final sentence and Figure S1: could the authors clearly state their conclusions when performing similar experiments on eukaryotic cells? For instance, do the authors think that lack of Mg2+ and divalent cations also distort 3D structures of eukaryotic ribosomal subunits?

Response: In the revised manuscript, the results for eukaryotic cells have been merged with E.coli’s as reviewer #2 comment on the significance of EDTA treatment. And the following sentence was added to the end of  the paragraph:

“The shifting of the subunit peaks in sucrose gradient profiles and the extended shapes of rRNA observed by cryoEM imaging indicated that extraction of Mg2+ from both prokaryotic and eukaryotic ribosomes by EDTA destroys the structure of ribosome. Following this, the destruction process was further studied using 30S subunits as a model with a series of cryoEM structures”

-p. 5/6, results paragraph “Structural distortion of 30S at low magnesium ion concentration”

Here again, justification of the use of a mutant version of DbpA to study the effect of magnesium depletion on 30S subunits is difficult to grasp. Why using this mutant if it is not involved in 30S subunit biogenesis? Furthermore, Fig. 2 displays gradient values that are different between fig 2a (20-40%) and panels 2b and 2c (10-50%); it is thus complicated to understand the correspondence between peaks. In figures 2a and 2b, how do the authors attribute the peaks to 30S, 45S and 50S? If DbpA is involved in 50S biogenesis, could the so-called 45S peak correspond to a diminished number of 50S subunits, while the so-called 50S peak actually be a full 70S ribosome? These remarks do not affect the strength of structural analysis presented in figure 2d and 2e. Could the authors clarify which of the “particles from the three datasets were extracted for further analysis”? Do they mean the three datasets that had were obtained with a concentration of 1mM Mg2+?

Response: Thanks for the reviewer’s comments. Please see the answers for the 2nd and 3rd comments about the use of DbpA which was now moved to the Materials and Method part. In this case, the Figure 2a was removed in the revised manuscript.

The 45S peak is acturally an immature 50S caused by DbpA R331A mutant, which was reported in a previous study (Gentry et al 2016). In the revised Figure 2, three datasets including 1, 2.5 and 10 mM Mg2+ was presented. And these three datasets were used for further analysis.

(Gentry, R.C., et al., Time course of large ribosomal subunit assembly in E. coli cells overexpressing a helicase inactive DbpA protein. RNA, 2016. 22(7): p. 1055-64.)

-p13, discussion chapter. The authors propose a very plausible model of 30S distortion caused by low Mg2+ concentration. How does this model relate to published cryo-EM studies of in vitro assembly of prokaryotic ribosomal subunits? Furthermore, many cryo-EM structures of eukaryotic pre-ribosomes show structural rearrangements of the forming pre-40S particle, with huge movements of rRNA helix h44 and head domains during nucleolar assembly steps (cf for instance the review from Barandun and Klinge, Curr Opin Struct Biol 2018, or scientific articles such as Kornprobst et al, Cell 2016), as well as cytoplasmic ones (see Heuer et al., eLife 2017, Scaiola et al, EMBO 2018, Shayan et al, Molecules 2020). Can the movements herein observed using low Mg2+ concentration be correlated with the structural rearrangements observed during 40S assembly?

Response: Thanks for the reviewer’s suggestion. In the discussion chapter, we have discussed the h28/h44 maturation accoding recent publications (Schedlbauer et al. 2022). Sentences like “with an h44 in the final position, is checked by RsgA, which means that the placing of the h44 helix is the last step in 30S maturation. Thus, it makes sense that h44 is the first part to be dissociated when the 30S structure becomes unstable.”. Also for h27 (Hori et al. 2021), “h27, under conditions of low Mg2+ concentration, also becomes more flexible than other helices”. Researches of eukaryotic pre-ribosomes were also discussed in the revised manuscript and the movements observed was correlated with the structural rearrangements during 30S/40S maturation.

Reviewer 2 Report

In this manuscript, the authors report the stability of ribosomes in different concentrations of Mg2+. They characterized the integrity of ribosomes in different concentrations of  Mg2+ mainly with cryo-EM. 

I have following comments based on their reported results:

1. What is the significance of the experiment with 10 mM EDTA?  As the author introduced in their introduction section, it is already known that the ribosomes are unstable at low concentration of Mg2+.  The physological concentration of the Mg2+ is between 1-5 mM. 

2. Previous and current studies all indicate that Mg2+ is critical in the stability and assembly of ribosome, but how it affect the structure? Do you see any Mg2+ binding sites in your structures ?

3. The authors claims that the map is reconstructed in the overall resolution of 3.3 angstrom. However, I feel the map quality is not as good as reported. For the quality of the map, please show some details of map/model fit, for both protein and RNA in the map.

4. Authors reported multiple maps in the paper, but I only found the data processing and FSC of one data set. Please show the FSC curve and sample density/map for all data sets presented in the paper. 

5. For the labeled residues in figure 4, how is the density quality? I only can see a overall shape of the density rather than detailed residues and RNA. Please add some enlarged density of them, such as A465-C470. 

6. In sucrose gradient profiles of figure 2/6, please add the numbers in Y axis. Besides the cryo-EM 2D images, do you have any other methods to validate the components on the 30s/50s/70s sample, such as SDS-PAGE? Is it possible that some components are missing during purification? 

Author Response

Thanks for the three reviewer’s suggestions, changes have been made in the revised manuscript, mainly including:

  1. The statement of DbpA was moved from the Results to the Materias and Methods part.
  2. Methods for growing yeast and human cells were added.
  3. Details of data processing for 2.5 mM and 10 mM Mg2+ particles were added.
  4. Figures S1 was merged with Figure 1 and the description and discussion were added to the first paragraph of the Results.
  5. DbpA related images were removed from Figure 2. Instead, the 2.5 mM Mg2+ data was merged from orignal Figure 6.
  6. Figure S9 (30S5mM) was merged with Figure S4 (30S10mM) and the data processing procedures, FSC, local resolutions were added.
  7. SDS-PAGE analysis was performed in Figure S8 and discussed in the last paragraph of Results.

Answers for the reviewers’ comments were listed below.

Reviewer #2

In this manuscript, the authors report the stability of ribosomes in different concentrations of Mg2+. They characterized the integrity of ribosomes in different concentrations of Mg2+ mainly with cryo-EM.

I have following comments based on their reported results:

  1. What is the significance of the experiment with 10 mM EDTA? As the author introduced in their introduction section, it is already known that the ribosomes are unstable at low concentration of Mg2+. The physological concentration of the Mg2+ is between 1-5 mM.

Response: Thanks for the reviewer’s comments. Athrough it is already known the ribosomes are unstable at low Mg2+ concentration, the details of de-structuration process was still unclear. To symmetrically study the structural changes at diferent concentration of Mg2+, the 10 mM EDTA treatment was first conducted as a extreme condition (0 mM Mg2+), followed by 0.5, 1, 2.5, 5, 10, 20 mM Mg2+ in our study.

  1. Previous and current studies all indicate that Mg2+ is critical in the stability and assembly of ribosome, but how it affect the structure? Do you see any Mg2+ binding sites in your structures ?

Response: Unfortunately, it is hard to see any Mg2+ binding sites in our structures since the resolutions were not high enough. And, probably due to the low concentraion system, Mg2+ become unstable in the structure, which causes invisible density for the ion.

  1. The authors claims that the map is reconstructed in the overall resolution of 3.3 angstrom. However, I feel the map quality is not as good as reported. For the quality of the map, please show some details of map/model fit, for both protein and RNA in the map.

Response: Thanks for the reviewer’s suggestion. Details of map/model fitting were provided in Figure S2c, d in the revised manuscript, for both protein and rRNA and slice view of the local resolution was showen in Figure S2b.

  1. Authors reported multiple maps in the paper, but I only found the data processing and FSC of one data set. Please show the FSC curve and sample density/map for all data sets presented in the paper.

Response: Thanks for the reviewer’s suggestion. All the data processing was now shown in Figure S2 and S3, and all FSC/sample map were added in the revised manuscript.

  1. For the labeled residues in figure 4, how is the density quality? I only can see a overall shape of the density rather than detailed residues and RNA. Please add some enlarged density of them, such as A465-C470.

Response: Since the local resolutions for the residues in Figure 4 are very low (please see Figure S2b), it is hard to add enlarged density of them. Only density shape of the whole helix could be shown as in Figure 4. That is why no details in hydrogen bond, electrostatic interaction was discussed in the manuscript.

  1. In sucrose gradient profiles of figure 2/6, please add the numbers in Y axis. Besides the cryo-EM 2D images, do you have any other methods to validate the components on the 30s/50s/70s sample, such as SDS-PAGE? Is it possible that some components are missing during purification?

Response: Thanks for the reviewer’s suggestion. Numbers in Y axis have been added in the revised Figures.

Yes, SDS-PAGE could be used to validate the components on the subunits sample. In the revised manuscript, a new figure with SDS-PAGE was shown and it indicated some protein bands were partially dissociated during purification under 0.5 and 1 mM Mg2+ compared 10 mM Mg2+ (Figure S9). Sentences describing these results were added to the revised manuscript as below:

“Analysis of the ribosomal proteins in 30S peak by SDS-PAGE showed several weakened bands at 0.5 and 1 mM Mg2+ compared to 10 mM Mg2+ (Figure S8). These could be the results of a weak binding to the rRNA at low Mg2+ concentration, followed by falling off from 30S when the concentration by centrifugation was applied”

Reviewer 3 Report

The authors present a work focusing on the ribosome 30S subunit structural distortion at different magnesium concentrations using cryoEM technology. Unlike most cryoEM projects aiming for high-resolution structures with optimized conditions to improve the homogeneity of the samples, the authors take a bold move to study disrupted structures in non-ideal conditions which potentially limits the ability of the technique to get enough resolution for structure interpretation, however, the study still provides scientific value for understanding the mechanism of ribosome assembly and contains certain novelty in the work. Since the map quality can be achieved is low, some analyses and conclusions should be carefully verified.  Please see below for the details. 

1. Figure 2e,  the overall structure of the 30S subunit at 1mM Mg2+ shows lower quality comparing the claimed resolution range and FSC. The author should provide 1-2 local region fitting figures to facilitate the resolution claim.  Furthermore, Figure S3b shows the local resolution distribution as the highest resolution at 3 Å. However, in most cases, the FSC indicates an overall resolution and is usually lower than the highest resolution. And the unfiltered unsharpened map is hard to verify the quality. The map also shows some sharp edges on the bottom, which is probably caused by a too-tight mask without a soft edge. To better visualize the local resolution distribution, a central section view of the map colored by local resolution distribution should also be included. 

2. The authors mentioned using multibody refinement to refine the 30S subunit head and body separately(Figure S5). They should first show the refinement results of 2 focus masked rigid bodies to confirm the alignment is reliable for both head and body regions. Then the movie interpretation could be trusted as valid distribution of flexibility. Otherwise, the 30S head is a relatively small domain, it is prone to overfitting without checking the map quality. 

3. The authors discussed the S12 missing density across different focused classification results. The maps in Figures d and e look reasonable and reliable to compare the S12 domain while the maps in Figure S7 could be over-interpreted with the limit of resolution. And the fact that a very small mask was used for the classification, the image alignment validation for classification could fail, only a strong signal or density should be trusted. The authors shouldn't be able to segment the density properly and assign the S12 domain correctly. Figure S7 and its claim should be rectified. 

4. cryoDRGN is a powerful tool to analyze the heterogeneity of the samples, but it's hard to qualify the 500 volumes the authors sampled from the latent space which involved density interpolation and it's dangerous to use with small domain interpretation (Figure 5). The author should provide references or instances in other similar works to justify that the practice in this work is properly following the conventions. 

Author Response

Thanks for the three reviewer’s suggestions, changes have been made in the revised manuscript, mainly including:

  1. The statement of DbpA was moved from the Results to the Materias and Methods part.
  2. Methods for growing yeast and human cells were added.
  3. Details of data processing for 2.5 mM and 10 mM Mg2+ particles were added.
  4. Figures S1 was merged with Figure 1 and the description and discussion were added to the first paragraph of the Results.
  5. DbpA related images were removed from Figure 2. Instead, the 2.5 mM Mg2+ data was merged from orignal Figure 6.
  6. Figure S9 (30S5mM) was merged with Figure S4 (30S10mM) and the data processing procedures, FSC, local resolutions were added.
  7. SDS-PAGE analysis was performed in Figure S8 and discussed in the last paragraph of Results.

Answers for the reviewers’ comments were listed below.

Reviewer #3

The authors present a work focusing on the ribosome 30S subunit structural distortion at different magnesium concentrations using cryoEM technology. Unlike most cryoEM projects aiming for high-resolution structures with optimized conditions to improve the homogeneity of the samples, the authors take a bold move to study disrupted structures in non-ideal conditions which potentially limits the ability of the technique to get enough resolution for structure interpretation, however, the study still provides scientific value for understanding the mechanism of ribosome assembly and contains certain novelty in the work. Since the map quality can be achieved is low, some analyses and conclusions should be carefully verified. Please see below for the details.

  1. Figure 2e, the overall structure of the 30S subunit at 1mM Mg2+ shows lower quality comparing the claimed resolution range and FSC. The author should provide 1-2 local region fitting figures to facilitate the resolution claim. Furthermore, Figure S3b shows the local resolution distribution as the highest resolution at 3 Å. However, in most cases, the FSC indicates an overall resolution and is usually lower than the highest resolution. And the unfiltered unsharpened map is hard to verify the quality. The map also shows some sharp edges on the bottom, which is probably caused by a too-tight mask without a soft edge. To better visualize the local resolution distribution, a central section view of the map colored by local resolution distribution should also be included.

Response: Thanks for the reviewer’s suggestion. Local region fitting figures for the rRNA and protein was provided in the revised Figure S2d and d. The local resolution map in Figure S3b (now Figure S2b) was remade for the sharpened map and the resolution range was changed to 2.8 to 7.2  Å. Also a slice map shown the resolution for the central region was provided.

  1. The authors mentioned using multibody refinement to refine the 30S subunit head and body separately(Figure S5). They should first show the refinement results of 2 focus masked rigid bodies to confirm the alignment is reliable for both head and body regions. Then the movie interpretation could be trusted as valid distribution of flexibility. Otherwise, the 30S head is a relatively small domain, it is prone to overfitting without checking the map quality.

Response: Thanks for the reviewer’s suggestion. The refinement results of the 2 focused/masked rigid bodies was shown in the revised Figure S5 (now Figure S4). The body part showed a better map quality that was expected, and the head map resolution is lower but showed a typical 30S head structure.

  1. The authors discussed the S12 missing density across different focused classification results. The maps in Figures d and e look reasonable and reliable to compare the S12 domain while the maps in Figure S7 could be over-interpreted with the limit of resolution. And the fact that a very small mask was used for the classification, the image alignment validation for classification could fail, only a strong signal or density should be trusted. The authors shouldn't be able to segment the density properly and assign the S12 domain correctly. Figure S7 and its claim should be rectified.

Response: Thanks for the reviewer’s comment. Mask on S12 protein was used for classification after refinement in Relion but without further alignment. As reviewer mentioned, it is hard to segment the density properly for S12 in Figure S7a, c, e, g, since they have very low resolution. Instead of color the assigned S12 map to red, we put the S12 pdb into the map to show the diverse of S12 occupancy in the revised Figure S7 (now Figure S6).

  1. cryoDRGN is a powerful tool to analyze the heterogeneity of the samples, but it's hard to qualify the 500 volumes the authors sampled from the latent space which involved density interpolation and it's dangerous to use with small domain interpretation (Figure 5). The author should provide references or instances in other similar works to justify that the practice in this work is properly following the conventions.

Response: In the cryoDRGN protocol (Kinman, L.F., et al. 2022), the assembling 50S ribosome dataset was used as an example, 96,478 particles was selected as good particles and send for 500-volumes analysis. In average about 200 paticles per volume. In our study, 440,474 particles for 500 volumes, about 880 per volume. So we think choosn of 500 volumes is proper. On the other hand, 200 volumes was actually used at the beginning of the analysis, and it showed the same results as 500 volumes. To be consistent with the cryoDRGN protocol, we use 500 volumes at the end. In the revised manuscript, this cryoDRGN protocol was cited in the correct positions.

(Kinman, L.F., et al., Uncovering structural ensembles from single-particle cryo-EM data using cryoDRGN. Nat Protoc, 2022.)

Round 2

Reviewer 2 Report

The author answered all the questions I asked and provided more data in figures. I think the manuscript is qualified for publication. 

Reviewer 3 Report

The authors managed to answer all the questions and solve the concerns the reviewers raised. The manuscript is in good shape to be published.